# Productivity, Nutrient Digestibility, Nitrogen Retention, and Meat Quality in Rabbits Fed Diets Supplemented with *Sida hermaphrodita*

**DOI:** 10.3390/ani9110901

**Published:** 2019-11-01

**Authors:** Cezary Purwin, Andrzej Gugołek, Janusz Strychalski, Maja Fijałkowska

**Affiliations:** 1Department of Animal Nutrition and Feed Science, University of Warmia and Mazury in Olsztyn, Oczapowskiego 5, 10-719 Olsztyn, Poland; purwin@uwm.edu.pl (C.P.); maja.fijalkowska@uwm.edu.pl (M.F.); 2Department of Fur-Bearing Animal Breeding and Game Management, University of Warmia and Mazury in Olsztyn, Oczapowskiego 5, 10-719 Olsztyn, Poland; janusz.strychalski@uwm.edu.pl

**Keywords:** rabbits, *Sida hermaphrodita*, production performance, nutrient digestibility, nitrogen retention, meat quality

## Abstract

**Simple Summary:**

Alfalfa is an important forage crop in rabbit nutrition. Despite the numerous advantages of alfalfa, efforts have been made to find its potential substitutes. Virginia fanpetals (*Sida hermaphrodita*), is yet another potential substitute for alfalfa in animal diets. The aim of this study was to evaluate the efficacy of dehydrated Virginia fanpetals meal as a substitute for dehydrated alfalfa meal in rabbit diets. The results of this study indicated that Virginia fanpetals meal can be included in rabbit diets at up to 20% as a substitute for alfalfa without compromising the production performance of animals, nutrient digestibility, nitrogen retention, carcass quality, or meat quality parameters.

**Abstract:**

Alfalfa (*Medicago sativa*) is an important forage crop in rabbit nutrition. Despite the numerous advantages of alfalfa, efforts have been made to find its potential substitutes. The aim of this study was to evaluate the efficacy of *Sida hermaphrodita* meal as a substitute for alfalfa meal in rabbit diets. The experiment was performed on 90 New Zealand White rabbits divided into three groups. DA group was fed a diet containing 20% dehydrated alfalfa. In the DA/DS group, rabbits received a diet containing 10% dehydrated alfalfa and 10% dehydrated Sida. The diet administered to the DS group contained 20% dehydrated Sida. The results of this study indicate that the dietary supplementation with Sida contributed to an increase in the final body weight of rabbits and improved the feed-conversion ratio. Experimental diets had no influence on nutrient digestibility, nitrogen retention, and selected carcass characteristics of rabbits, except for the proportion of the hind part. The content of dry matter, total protein, and monounsaturated fatty acids in the hind leg muscles of rabbits was higher in the DA group than in the experimental groups. The concentrations of saturated fatty acids were higher in the tissues of animals fed diets supplemented with Sida.

## 1. Introduction

Alfalfa (*Medicago sativa*) is an important forage crop in rabbit nutrition. Both alfalfa hay and dehydrated alfalfa meal are used as ingredients of complete diets. Alfalfa is a rich source of fiber and protein in rabbit diets [1,2,3]. According to the literature, the alfalfa content of rabbit diets should range from 20% up to even 96%, whereas practical diets usually contain 30% to 40% alfalfa [2,4,5]. Despite the numerous advantages of alfalfa, efforts have been made to find its potential substitutes, particularly in tropical regions where alfalfa is not widely grown. There is also a need for cheaper alternative feed ingredients. The list of potential alfalfa substitutes that have been tested in recent years includes amaranth meal (*Amaranthus hypochondriachus*), bambara groundnut (*Vigna subterranean*), perennial peanut (*Arachis glabrata*), artichoke leaves (*Cynara cardunculus*), bermudagrass (*Cynodon dactylon*), berseem hay (*Trifolium alexandrinum*), birdsfoot trefoil (*Lotus corniculatus*), pigeon pea leaves (*Cajanus cajan*), kudzu (*Pueraria* spp.), orchard grass (*Dactylis glomerata*), water hyacinth (*Eichhornia crassipes*), moringa leaves (*Moringa oleifera*), and many other [6,7,8,9,10,11,12,13,14,15].

Virginia fanpetals (*Sida hermaphrodita*), a member of the Mallow (*Malvaceae*) family native to North America, is yet another potential substitute for alfalfa in animal diets. Virginia fanpetals has many advantages. Due to its specific biochemical composition, Sida in the green stage can be used as a highly nutritious feed with a protein content of up to 30%. When harvested in the dry stage, the species has various applications in the bioenergy, insulation, and cellulose-paper industries. When harvested in the green stage, it can be used in the pharmaceutical, animal feed, honey, and biogas industries. Sida is also a profitable energy crop considered a future feedstock for biofuel production [16,17,18,19].

*Sida hermaphrodita* has already been fed to cattle and pigs [20,21,22], and the promising results have promoted research into its use in other livestock species. Another plant of the family *Malvaceae, Sida acuta,* has been successfully tested in rabbits [23,24,25].

The objective of this study was to evaluate the efficacy of dehydrated *Sida hermaphrodita* meal as a substitute for dehydrated alfalfa (*Medicago sativa*) meal in diets for growing meat-type rabbits.

## 2. Materials and Methods

The animal protocol and the number of animals used in this study were consistent with regulations of the Local Institutional Animal Care and Use Committee (21/2018 Olsztyn, Poland), and the study was carried out in accordance with EU Directive 2010/63/EU on the protection of animals used for scientific purposes [26].

### 2.1. Animals and Housing

The experimental animals were 90 New Zealand White rabbits (45 females and 45 males). The rabbits were randomly allocated to three groups. When the experiment began, the animals were 35 days old and had average body weight (BW) of 908.97 ± 104.11 g (mean ± SD). The animals were 90 days old when the experiment ended. The experiment was performed between September and November, in a separate facility on a rabbit farm. All rabbits were kept in wire-mesh flat-deck cages measuring 0.5 × 0.6 × 0.4 m (two animals per cage until 45 days of age, then one animal per cage until the end of the experiment). Rabbits had ad libitum access to feed served once a day via automatic feeders and water from nipple drinkers. The animals were housed under standard conditions with a temperature of 16–18 °C, relative air humidity of 60–75%, forced room ventilation, and a controlled photoperiod (12 h light with intensity of 25 lx and 12 h dark).

### 2.2. Diets and Experimental Procedures

Control group (DA) rabbits were fed a diet containing 20% dehydrated alfalfa meal. In the first experimental group (DA/DS), rabbits received a diet containing 10% dehydrated alfalfa and 10% dehydrated Sida meal (DS). The diet administered to the second experimental group (DS) contained 20% dehydrated Sida meal. The ingredients of diets are presented in Table 1, whereas the chemical composition of diets and experimental factors are presented in Table 2. The fatty acid profiles of dehydrated alfalfa meal and dehydrated Sida meal (% total fatty acid pool) are presented in Table 3.

All diets were isonitrogenous and their nutritional value corresponded to the requirements of growing meat-type rabbits [2]. Dehydrated alfalfa meal was purchased from a local commercial feed mill. Dehydrated Sida meal was prepared in the laboratory of the Department of Animal Nutrition and Feed Science, University of Warmia and Mazury in Olsztyn. Sida biomass was collected on a plantation in the fifth year of its productive life, located in northern Poland on sandy soil fertilized with N_80_ P_20_K_60_ kg·ha^−1^. The plants were cut before budding, at a height of 30 cm, they were chopped with an electric chaff cutter and dried at a temperature of 60 °C in BINDER dryers. After 24 h, dried biomass was ground in a mill (ZM 200, Retsch, Haan, Germany) to a 2-mm particle size. The feed was mixed in the Zuptor 300 horizontal feed mixer (Gostyń, Poland) and formulated in the MAGNUS 100 compact feed pelletizer (Ożarów Poland) with a matrix diameter of 100 mm and a pellet diameter of 6 mm.

During the experiment, the rabbits were weighed on an electronic scale within an accuracy of 1 g, and their BW were determined at the beginning of the experiment (35 days of age) and at the end of the experiment (90 days of age). Average daily weight gains, total feed intake and the feed conversion ratio (FCR) were calculated.

A digestion and balance trial was performed on 10 rabbits (5 males and 5 females) selected randomly from each group. The animals were 45 days old when the digestion and balance trial began, and 65 days old when the trial ended. They were placed individually in metabolic cages fitted with a system for quantitative collection of feces and urine. During the 10-day adaptation period that preceded the experiment proper, the rabbits were allowed to adapt to the new environment and feed. Pelleted feed in the amount of 150 g was served once a day at 10:00. The rabbits had free access to drinking water. Non-ingested feed residues and feces were collected every day and weighed within an accuracy of 1 g. The collected feces were frozen, and feces and feed samples were dried and ground. Urine was preserved with 20% sulfuric acid to calculate the total volume of the collected urine at the end of the experiment. The chemical composition and energy value of feed samples were determined.

The balance method used in studies of the type supports the calculation of nutrient and energy digestibility coefficients (DC) and nitrogen (N) retention. Nutrient digestibility was calculated from the following formula: DC = (a − b)/a × 100%, where a is the nutrient content of feed, and b is the nutrient content of feces.

At the end of the feeding trial, the animals were fasted for 24 h and sacrificed according to the standard guidelines for euthanizing experimental animals. The carcasses were skinned and eviscerated. The head was dissected along the occipital joint, the forepart was dissected between the 7th and 8th thoracic vertebrae, and the loin was dissected between the 6th and 7th lumbar vertebrae. The hind part with the perisacral area and hind legs was the remaining part of the carcass after dissection [27]. The following slaughter performance data were collected: Pre-slaughter weight, carcass weight with and without the head, and dressing percentage with and without the head. Dressing percentages with (I) and without the head (II) were calculated according to the following formulas: DP I = carcass weight with the head/slaughter weight × 100%, DP II = carcass weight without the head/slaughter weight × 100%. The proportion of the most valuable carcass cuts, i.e., the forepart, loin, and the hind part, was expressed in “g”.

### 2.3. Analytical Methods

Hind leg muscles were collected from 5 males and 5 females from each group for analyses of the chemical composition and fatty acid profile of meat after 24 h of chilling at +4 °C. Feed and feces samples were also subjected to chemical analyses. The content of dry matter, crude ash, total protein, ether extract, acid detergent fiber (ADF), and acid detergent lignin (ADL) was determined by standard methods [28]. Neutral detergent fiber (NDF), ADF and ADL were estimated in the FOSS TECATOR Fibertec 2010 System. NDF was determined according to the procedure proposed by Van Soest et al. [29]. The levels of amino acids in diets were determined using the Biochrom 20 plus amino acid analyzer and Biochrom amino acid analysis reagents (Biochrom Ltd., Cambridge, England). Gross energy content was determined using a bomb calorimeter (IKA^®^ C2000 basic, Germany, Staufen). To determine fatty acid composition, all fat samples were methylated by the modified Peisker method [30] (1.5 cm^3^ of a methanol:chloroform:concentrated sulfuric acid mixture, 100:100:1 v/v, was added to ca. 150 μL fat, thermostat −80 °C, 3 h), and fatty acid methyl esters were obtained. Fatty acids were separated and determined by gas chromatography: VARIAN CP–3800 gas chromatograph-Netherlands, flame-ionization detector (FID), capillary column (length −50 m, Φ = 0.25 mm, film d = 0.25 μm), split injector, split ratio 50:1, 1 μL sample, detector temperature −250 °C, injector temperature −225 °C, column temperature −200 °C, carrier gas—helium, flow rate −1.2 cm^3^/min. Fatty acids were identified by comparing the retention times of individual fatty acid methyl ester standards (Sigma-Aldrich, St. Louis, MO, USA) and the retention times of peaks in the analyzed samples. The relative content of each fatty acid was expressed as a percent of the total peak area of all fatty acids in the sample. The concentrations of saturated fatty acids (SFAs), monounsaturated fatty acids (MUFAs) and polyunsaturated fatty acids (PUFAs) were calculated.

### 2.4. Statistical Analyses

Data were expressed as means ± standard deviation of the mean (SD). The results were analyzed statistically by one-way analysis of variance (ANOVA), and the significance of differences among groups was determined by Duncan’s multiple range test at *p* equal to or less than 0.05 and 0.001. All calculations were performed using Statistica 12.0 software [31].

## 3. Results

The initial and final BW of rabbits from the control group and experimental groups did not differ significantly (Table 4). However, the final BW of rabbits fed Sida-supplemented diets were higher than those of animals fed alfalfa-based diets: Group DA/DS (2561.02 g) and group DS (2481.40 g) vs. group DA (2392.08 g). Average daily weight gains ranged from 26.97 g to 29.96 g, and no significant differences were found between groups, although daily gains tended to increase in rabbits receiving dehydrated Sida meal, similarly to BW values. Feed intake was similar in all groups, whereas total BW gain was significantly higher in group DA/DS than in group DA (1.65 vs. 1.48 kg). The FCR was lower in the experimental groups than in the control group, and it reached the lowest level in group DA/DS −4.05 kg.

Nutrient digestibility (Table 5) was similar in all groups. No significant differences in nitrogen retention were observed between groups. However, nitrogen retention (%) tended to be higher in the experimental groups, and the highest daily nitrogen retention (g) as well as nitrogen retention (%) with respect to nitrogen uptake and digestion were noted in group DA/DS fed a diet containing 10% dehydrated alfalfa and 10% dehydrated Sida meal.

Selected carcass characteristics, presented in Table 6, did not differ significantly between groups, except for the proportion of the hind part which was significantly higher in group DA/DS (442.43 g) than in group DA (390.78 g). In general, carcass weight, dressing percentage, and the weights of carcass cuts were lower in the control group (DA) than in the experimental groups.

The content of dry matter and total protein in hind leg muscles was significantly higher in the control group than in the experimental groups (Table 7), whereas ether extract content was lowest in group DA. Crude ash content was highly similar in all groups, ranging from 1.21% to 1.22%.

The fatty acid profiles of hind leg muscles are shown in Table 8. The concentrations of SFAs were higher in groups fed Sida-supplemented diets, and MUFA levels were higher in the control group (DA). An analysis of individual fatty acids revealed that the content of C_14:0_ and C_16:0_ was higher in group DS than in group DA (20% dehydrated alfalfa meal), the content of C_18:0_ was higher in groups DA/DS and DS than in group DA, and the content of C_16:1_, C_17:1_ and C_18:1_ was higher in group DA (control) than in the experimental groups. No significant differences in PUFA levels were found between groups.

Mortality or serious diseases symptoms that could affect the results of this study were not observed during the experiment.

## 4. Discussion

The performance parameters of rabbits from all groups (Table 4) were within normal limits for growing meat-type rabbits of medium-sized breeds raised in Europe [5,27,32]. However, only a few studies have investigated the efficacy of *Sida hermaphrodita* in livestock nutrition, and there appears to be no published information on rabbits. Therefore, it was difficult to compare the results of this study with the findings of other authors. In general, fresh and dehydrated alfalfa is regarded as a valuable feed ingredient for rabbits [14,33,34,35,36]. According to some authors, increasing dietary inclusion levels of alfalfa contributes to production results in rabbits [37,38]. In this study, the performance parameters of experimental animals (groups DA/DS and DS) were more desirable, compared with the control group fed alfalfa-based diets, which suggests that Sida exerted a beneficial influence on rabbits. However, the observed positive effects cannot be attributed to the chemical composition of feed because the levels of nutrients and selected amino acids in dehydrated alfalfa meal and dehydrated Sida meal were similar (Table 2). Nevertheless, it appears that *Sida hermaphrodita,* similarly to other species of the genus Sida (*Malvaceae*), *Sida acuta, Sida cordifolia, Sida spinosa, Sida rhombifolia,* and *Sida veronicaefolia,* has health-promoting properties which positively affect the health status and, indirectly, productivity of animals [39]. However, further research is needed to validate this hypothesis. It should be noted that studies investigating another plant species of the genus *Malvaceae, Sida acuta,* revealed that experimental diets had no adverse effects on performance or overall health in rabbits [23,24,25].

The values of nutrient digestibility coefficients and nitrogen retention, calculated in this experiment, are typical of growing broiler rabbits [5,32]. Despite an absence of significant differences between group means, Table 5 data correspond to the performance parameters presented in Table 4 because more desirable trends were noted in the experimental groups receiving dehydrated Sida meal. The effect of *Sida hermaphrodita* on nutrient digestibility in rabbits or other animal species has not been examined to date, except for a study investigating gilts, which revealed an increase in protein and fiber digestibility with increasing levels of Sida meal (up to 10%) in the ration [20]. However, the results of the above study cannot be extrapolated to rabbits due to considerable physiological differences between the analyzed species. In a study of domestic and wild rabbits [36], alfalfa-supplemented diets increased or did not affect nutrient digestibility. Sun et al. [14], who analyzed rabbits fed diets containing 30% alfalfa meal, demonstrated that protein digestibility reached 61%, which is consistent with the values determined in our study. The cited authors also observed an increase in dry matter digestibility associated with the dietary inclusion levels of alfalfa.

The lower average weights of carcass and valuable cuts in group DA resulted from lower slaughter weight of rabbits (Table 6). The values of dressing percentage without the head (51.44–53.94%) and with the head (46.06–48.43%) are similar to those reported in previous studies [27,32]. The values of carcass quality parameters are related to the growth performance of rabbits (Table 4), nutrient digestibility, and nitrogen retention (Table 5).

The proximate chemical composition of meat (Table 7) was typical of broiler rabbits slaughtered at 90 days of age [27,40]. In a study by Kowalska et al. [40], the content of total protein and crude ash in rabbit meat ranged from 21.78% to 23.41%, and from 1.15% to 1.17%, respectively. In the work of Daszkiewicz et al. [27], the dry matter and total protein content of rabbit meat reached 23.88–24.27% and 22.18–22.78%, respectively. The values reported by the above authors are similar to those determined in the current study. The increase in the concentrations of dry matter and total protein in the hind leg muscles of rabbits, observed in this experiment, is difficult to explain because it did not correspond to the similar chemical composition and fatty acid profiles of alfalfa, Sida, and the respective diets (Table 2) or higher nitrogen retention in group DA/DS (Table 5). According to previous research, the fatty acid profile of diets fed to animals affects the fatty acid profile of their meat. Lin et al. [41] demonstrated that the fatty acids of dietary fats may greatly influence the fatty acid composition of adipose tissue in rabbits.

The results of numerous studies show that feeding high levels of alfalfa meal has a positive effect on the quality of rabbit meat. It may increase the concentrations of total essential amino acids, taste-related amino acids, and alpha-linolenic acid, thus improving the nutritional and organoleptic quality of meat [38,42]. Capra et al. [33] noted an increase in linolenic acid content and an improvement in the n-6/n-3 PUFA ratio in the tissues of rabbits fed alfalfa-supplemented diets. Dal Bosco et al. [34] attempted to improve the quality of rabbit meat by increasing the levels of natural bioactive compounds through providing fresh alfalfa to rabbits. The cited authors found that the dietary supplementation with fresh alfalfa led to an increase in the concentrations of stearic, linolenic, eicosatrienoic, eicosapentaenoic, docosapentaenoic, and docosahexaenoic acids, and total PUFAs in meat, whereas MUFA content was considerably lower, in contrast to our study. Similar but less pronounced effects of alfalfa on the quality of rabbit meat were reported by Dal Bosco et al. [35]. In the cited study by Dal Bosco et al. [34], the proportions of fatty acid groups in the longissimus lumborum muscle of rabbits fed diets with increased alfalfa content were as follows: SFAs 41.9%, MUFAs 22.6%, and PUFAs 35.9%. Bianchi et al. [42] found that the fatty acid composition of the hind leg muscles of rabbits receiving 25% dehydrated alfalfa was as follows: SFAs 39.81–41.15%, MUFAs 28.05–30.01%, and PUFAs 23.69–23.75%. Thus, the fatty acid profile of hind leg muscles in rabbits from group DA (Table 8) can be considered typical of rabbits fed diets supplemented with dehydrated alfalfa. The increase in SFA levels and the decrease in MUFA content, observed in the experimental groups, were most likely due to their presence in dehydrated Sida meal (Table 3).

## 5. Conclusions

The results of this study indicate that *Sida hermaphrodita* can be included in rabbit diets at up to 20% as a substitute for alfalfa without compromising the production performance of animals, nutrient digestibility, nitrogen retention, carcass quality, or meat quality parameters.

## Figures and Tables

**Table 1 animals-09-00901-t001:** Composition of diets (%).

Components	Diet		
	DA	DA/DS	DS
Soybean meal	10.0	10.0	10.0
Dehydrated alfalfa meal	20.0	10.0	0.0
Dehydrated Sida meal	0.0	10.0	20.0
Wheat bran	42.0	42.0	42.0
Rapeseed meal	6.0	6.0	6.0
Corn DDGS *	6.0	6.0	6.0
Arbocel **	6.0	6.0	6.0
Dried beet pulp	5.0	5.0	5.0
Dried brewer’s yeast	1.0	1.0	1.0
Whey powder	1.0	1.0	1.0
Sodium chloride (NaCl)	0.2	0.2	0.2
Lime	1.3	1.3	1.3
Phosphate	0.5	0.5	0.5
Mineral-vitamin premix ***	1.0	1.0	1.0
Total	100.0	100.0	100.0

DA—Control group (DA) rabbits were fed a diet containing dehydrated alfalfa mea; DS—experimental group (DS) contained dehydrated Sida meal. * Dried distilled grains with solubles. ** Crude fiber concentrate. *** Composition of mineral-vitamin premix (1 kg): Vitamin A—3,500,000 IU; vitamin D_3_—200,000 IU; vitamin E—28,000 mg; vitamin K_3_—200 mg; vitamin B_1_—1500 mg; vitamin B_2_—2800 mg; vitamin B_6_—2800 mg; vitamin B_12_—20,000 mcg; folic acid—200 mg; niacin—10,000 mg; biotin—200,000 mcg; calcium pantothenate—7000 mg; choline—30,000 mg; Fe—17,000 mg; Zn—2000 mg; Mn—1000 mg; Cu (copper sulfate × 5H_2_O, 24.5%)—800 mg; Co—1000 mg; I—100 mg; Ca—150 g; P—100 g.

**Table 2 animals-09-00901-t002:** Chemical composition of diets and experimental factors (%).

	Diet			Dehydrated Alfalfa Meal	Dehydrated Sida Meal
	DA	DA/DS	DS		
Dry matter	88.78	88.65	88.59	92.58	91.41
Crude ash	7.23	7.25	7.29	8.29	10.59
Organic matter	81.55	81.40	81.30	84.29	80.82
Total protein	18.42	18.36	18.28	19.24	18.66
Crude fat	3.31	3.33	3.37	1.63	1.89
NDF	28.98	29.08	29.34	31.71	31.92
ADF	16.46	17.18	17.89	22.65	29.94
ADL	5.12	5.45	5.72	3.22	6.40
Lysine	0.92	0.92	0.92	0.94	0.91
Methionine + cystine	0.88	0.88	0.88	0.52	0.52
Threonine	0.81	0.79	0.78	0.83	0.66
Thrypophan	0.21	0.21	0.20	0.33	0.28
Gross energy [MJ/kg]	16.48	16.49	16.51	16.18	16.29

**Table 3 animals-09-00901-t003:** Fatty acid profiles of dehydrated alfalfa meal and dehydrated Sida meal (% total fatty acid pool).

Fatty Acid.	Dehydrated Alfalfa Meal	Dehydrated Sida Meal
C_12:0_	0.32	0.21
C_14:0_	1.14	1.54
C_15:0_	1.03	1.14
C_16:0_	28.94	31.60
C_16:1_	1.28	1.01
C_17:0_	0.64	0.85
C_17:1_	0.68	0.30
C_18:0_	6.01	8.56
C_18:1_	11.30	6.38
C_18:2_	20.72	20.17
C_18:3_	25.67	26.36
C_20:0_	1.12	0.99
C_20:1_	0.32	0.41
C_20:2_	0.62	0.15
C_20:4_	0.21	0.31
C_22:0_	0.00	0.02
SFAs	39.20	44.91
MUFAs	13.58	8.10
PUFAs	47.22	46.99

SFAs, saturated fatty acids; MUFAs, monounsaturated fatty acids; PUFAs, polyunsaturated fatty acids.

**Table 4 animals-09-00901-t004:** Production performance of rabbits (mean ± SD; n = 30).

Specification	Group			*p*-Value
	DA	DA/DS	DS	
Body weight, 35 days (g)	908.36 ± 90.98	913.10 ± 121.94	905.44 ± 99.40	0.616
Body weight, 90 days (g)	2392.08 ± 181.16	2561.02 ± 211.40	2481.40 ± 217.79	0.089
Daily body weight gain, 35–90 days (g)	26.97 ± 1.13	29.96 ± 0.77	28.65 ± 0.76	0.198
Feed intake (kg)	6.67 ± 2.08	6.69 ± 1.81	6.56 ± 2.27	0.161
Total body weight gain (kg)	1.48 ± 0.22 ^b^	1.65 ± 0.32 ^a^	1.58 ± 0.18	0.048
Feed conversion ratio (kg/kg)	4.51 ± 0.05 ^a^	4.05 ± 0.06 ^b^	4.15 ± 0.03 ^b^	0.025

^a,b^ within rows, values with different letters are significantly different (*p* < 0.05).

**Table 5 animals-09-00901-t005:** Nutrient digestibility and nitrogen utilization in rabbits (mean ± SD; n = 10).

Specification	Group			*p*-Value
	DA	DA/DS	DS	
Digestibility coefficients (%):				
Dry matter	70.78 ± 4.01	70.75 ± 8.15	69.66 ± 5.07	0.785
Organic matter	62.19 ± 4.07	62.33 ± 8.02	60.51 ± 5.11	0.844
Total protein	71.77 ± 2.77	72.29 ± 6.03	71.70 ± 388	0.822
Crude fat	82.43 ± 8.69	86.35 ± 3.28	84.86 ± 10.00	0.799
Neutral detergent fiber	36.20 ± 7.31	36.64 ± 13.25	35.31 ± 11.18	0.926
Acid detergent fiber	25.19 ± 10.41	26.94 ± 14.23	24.92 ± 10.56	0.843
Acid detergent lignin	32.94 ± 17.53	35.82 ± 17.10	35.91 ± 7.40	0.512
Gross energy	61.11 ± 5.26	61.76 ± 8.79	61.63 ± 5.16	0.984
Daily nitrogen balance (g/rabbit):				
Uptake	4.49 ± 0.58	4.32 ± 0.90	4.55 ± 0.33	0.682
Excretion with feces	1.29 ± 0.24	1.12 ± 0.33	1.25 ± 0.18	0.124
Excretion with urine	2.12 ± 0.28	1.93 ± 0.41	2.10 ± 0.32	0.505
Digestion	3.20 ± 0.44	3.20 ± 0.58	3.30 ± 0.28	1.000
Retention	1.09 ± 0.33	1.27 ± 0.36	1.20 ± 0.17	0.409
Retention with respect to nitrogen (%):				
Uptake	24.13 ± 7.14	29.69 ± 5.94	26.37 ± 4.01	0.133
Digestion	33.50 ± 8.25	39.65 ± 7.17	36.36 ± 5.69	0.241

Nonsignificant differences.

**Table 6 animals-09-00901-t006:** Carcass characteristics of rabbits (mean ± SD; n = 30).

	Group			*p*-Value
	DA	DA/DS	DS	
Body weight (g)	2392.08 ± 181.16	2561.02 ± 211.40	2481.40 ± 217.79	0.089
Carcass weight with head (g)	1230.56 ± 128.14	1372.29 ± 118.14	1338.44 ± 122.19	0.071
Carcass weight without head (g)	1101.78 ± 112.21	1236.43 ± 103.02	1201.78 ± 113.72	0.093
Dressing percentage with head (%)	51.44 ± 2.81	53.58 ± 3.50	53.94 ± 2.26	0.312
Dressing percentage without head (%)	46.06 ± 2.31	48.28 ± 3.15	48.43 ± 2.03	0.253
Forepart (g)	426.56 ± 51.92	483.14 ± 54.88	471.89 ± 46.69	0.076
Loin (g)	284.44 ± 42.69	310.86 ± 33.49	304.78 ± 35.75	0.346
Hind part (g)	390.78 ± 39.79 ^b^	442.43 ± 34.94 ^a^	425.11 ± 41.11	0.041

^a,b^ within rows, values with different letters are significantly different (*p* < 0.05).

**Table 7 animals-09-00901-t007:** Proximate chemical composition of hind leg muscles in rabbits (%; mean ± SD; n = 10).

	Group			*p*-Value
	DA	DA/DS	DS	
Dry matter	25.41 ± 0.61 ^A^	24.49 ± 0.46 ^B^	24.24 ± 0.60 ^B^	0.000
Crude ash	1.22 ± 0.03	1.21 ± 0.03	1.21 ± 0.02	0.641
Total protein	23.02 ± 0.62 ^a^	22.58 ± 0.46	22.33 ± 0.19 ^b^	0.025
Ether extract	0.9 7± 0.37	1.19 ± 0.35	1.12 ± 0.59	0.571

^a,b^ within rows, values with different letters are significantly different (*p* < 0.05); ^A,B^ within rows, values with different letters are significantly different (*p* < 0.001).

**Table 8 animals-09-00901-t008:** Fatty acid profiles in the hind leg muscles of rabbits (% total fatty acid pool; mean ± SD; n = 10).

Fatty Acid		Group			*p*-Value
		DA	DA/DS	DS	
C_12:0_	Lauric acid	0.13 ± 0.04	0.14 ± 0.09	0.16 ± 0.07	0737
C_14:0_	Myristic acid	2.02 ± 0.34 ^B^	2.37 ± 0.34	2.74 ± 0.44 ^A^	0.003
C_14:1_	Myristoleic acid	0.13 ± 0.05	0.10 ± 0.03	0.13 ± 0.05	0.279
C_15:0_	Pentadecanoic acid	0.57 ± 0.10	0.52 ± 0.11	0.54 ± 0.07	0.590
C_16:0_	Palmitic acid	27.65 ± 3.41 ^b^	29.80 ± 2.94	31.69 ± 2.21 ^a^	0.040
C_16:1_	Palmitoleic acid	2.73 ± 0.77 ^A,a^	1.64 ± 0.43 ^B^	1.87 ± 0.34 ^b^	0.001
C_17:0_	Heptadecanoic (margaric) acid	0.79 ± 0.18	0.85 ± 0.22	0.84 ± 0.13	0.752
C_17:1_	Margaroleic acid	0.43 ± 0.07 ^A^	0.28 ± 0.05 ^B^	0.30 ± 0.06 ^B^	0.000
C_18:0_	Stearic acid	10.17 ± 1.58 ^B^	13.21 ± 1.55 ^A^	12.65 ± 1.0 ^A^	0.000
C_18:1_	Oleic acid	24.73 ± 1.36 ^A^	21.23 ± 1.62 ^B^	20.37 ± 1.1 ^B^	0.000
C_18:2_	Linoleic acid	22.13 ± 3.74	21.99 ± 3.78	20.85 ± 2.14	0.725
C_18:3_	α-linolenic acid	4.60 ± 2.09	4.11 ± 1.48	4.01 ± 1.16	0.744
C_20:0_	Arachidic (eicosanoic) acid	0.24 ± 0.06	0.22 ± 0.02	0.23 ± 0.04	0.472
C_20:1_	Gadoleic acid	0.36 ± 0.09	0.32 ± 0.08	0.33 ± 0.09	0.572
C_20:2_	Eicosadienoic acid	0.34 ± 0.07	0.35 ± 0.10	0.31 ± 0.08	0.691
C_20:4_	Arachidonic acid	2.08 ± 0.65	2.05 ± 0.74	2.13 ± 0.51	0.975
C_20:5_	Eicosapentaenoic acid (EPA)	0.02 ± 0.01	0.04 ± 0.08	0.04 ± 0.05	0.596
C_22:0_	Behenic acid	0.27 ± 0.08	0.23 ± 0.08	0.23 ± 0.05	0.467
C_22:5_	Docosapentaenoic acid (DPA)	0.51 ± 0.25	0.45 ± 0.22	0.50 ± 0.23	0.839
C_22:6_	Docosahexaenoic acid (DHA)	0.12 ± 0.03	0.09 ± 0.04	0.08 ± 0.05	0.263
SFAs		41.48 ± 5.17 ^b^	47.35 ± 4.49 ^a^	49.08 ± 3.49 ^a^	0.010
MUFAs		28.38 ± 1.88 ^A^	23.57 ± 1.64 ^B^	23.00 ± 0.9 ^B^	0.000
PUFAs		29.78 ± 5.68	29.08 ± 5.56	27.92 ± 3.52	0.773

^A,B^ within rows, values with different letters are significantly different (*p* < 0.001); ^a,b^ within rows, values with different letters are significantly different (*p* < 0.05); SFAs, saturated fatty acids; MUFAs, monounsaturated fatty acids; PUFAs, polyunsaturated fatty acids.

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
