# Peer review of "Productivity, Nutrient Digestibility, Nitrogen Retention, and Meat Quality in Rabbits Fed Diets Supplemented with Sida hermaphrodita"

_animals, 2019, doi:10.3390/ani9110901_

Round 1

Reviewer 1 Report

In my opinion this is a well written paper that deals with a very current and interesting topic in rabbit farming systems.

The methods and numbers of samples used are correct.

The discussion of the results is exhaustive and clear.

The only doubt I have is that the Authors carried out qualitative analyses at the level of hind leg and not on Longissimus dorsi muscle where most bibliographic data are present; but it is only my perplexity that does not affect the average quality of the manuscript.

Author Response

Thank you to the Reviewer for the positive evaluation of our manuscript.

Similar to our research, some articles considering influence of nutrition on hind leg meat can be found, e.g. Meat Science 2011, 87(1): 40-45; Animal Science Papers and Reports 2009, 27(2): 139-148, Annals of Animal Science 2013, 13(3): 571–585. So, as written by the Reviewer, it does not affect the quality of the manuscript. On the other hand, we agree with the Reviewer that for analyzes Longissimus dorsi muscle is used more often than hind leg. Thank you that you drew attention to it, in the future we will carry out analyzes primarily on the Longissimus dorsi muscle.

Reviewer 2 Report

Review of the manuscript animals-598152.

Dear Authors,

I kindly accepted the invitation of review on the paper: “Productivity, nutrient digestibility, nitrogen retention and meat quality in rabbits fed diets supplemented with Sida hermaphrodita”, and present herein the review report.  

The manuscript is very well written. The methodology, study and discussion are comprehensive, logical and reasonable. Substantial amount of analytical procedures has been used in the study, and results are described and analyzed in very professional manner. In spite of limited reference, especially with no similar data that has been published on studies with rabbits, the manuscript provides rational explanations and discussion.

I have only two suggestions.:

I advise to remove the gross energy value from table 2 because it not represents the nutritive characteristic of diet or test ingredient. Gross energy represents the amount of heat measured from complete combustion of the material in calorimeter bomb (furnace) and this value would provide just a number needed to calculate the energy digestibility. Edits:

Lines: 179-183 and 201-207: Pleas correct- the text is edited as a footnote.

Table 4: Daily body weight gain, 35-90 days; not Daily body weight gain, 35-901 days

Table 8: Arachidic acid; not: Aarachidic

I would recommend the manuscript for publication.

Best regards.

Author Response

Dear Reviewer,

Thank you for a thorough review of our manuscript.

Table 2 presents the analysis of basic chemical composition and energy obtained as a result of laboratory analyzes, including GE. In our opinion the parameter (GE) is very useful to present in table 2 because it was a base to count  the digestibility coefficients of energy – table 5 . Please pay attention GE has been presented in other publications. I hope you will agree with above.

Lines: 179-183 and 201-207  - the text was corrected according to the reviewer's suggestions.

Tables 4 and 8 were corrected in accordance with the reviewer's comments: on 35-90 days and arachidic acid.

Reviewer 3 Report

I would like to congratulate the Authors on the fine manuscript.

Just a few, minor suggestion:

L117: replace 'metabolism' cages with 'metabolic' cages

L118: correct version would be: 'During 10-days adaptation...'

L180: in the footnote to Table 4, please rephrase 'highly similar', which is not informative (just 'similar' would be fine...)

L223: the phrase 'production performance' feels slightly odd, please consider rephrasing

Author Response

Dear Reviewer,

We would like to thank for a kind review of the manuscript and the valuable suggestions. Below find, please, the improvements:

Line 117: replaced 'metabolism' cages into 'metabolic' cages

Line 118: changed to 'During 10-days adaptation' as suggested

Line 180: under Table 4, replaced 'highly similar' into 'similar'

Line: 223: the term 'production performance' has been replaced by ,production results’

This manuscript is a resubmission of an earlier submission. The following is a list of the peer review reports and author responses from that submission.